# Technology-based group exercise interventions for people living with dementia or mild cognitive impairment: A scoping review

Lillian Hung[1,2]*, Juyong Park[3], Hannah Levine[4], David Call[5], Diane Celeste[6], Dierdre Lacativa[6], Betty Riley[6], Nathanul Riley[5], Yong Zhao[2]

1 School of Nursing, University of British Columbia, Vancouver, British Columbia, Canada, 2 IDEA Lab, University of British Columbia, Vancouver, British Columbia, Canada, 3 Phyllis & Harvey Sandler School of Social Work, College of Social Work & Criminal Justice, Florida Atlantic University, Boca Raton, Florida, United States of America, 4 Charles E. Schmidt College of Medicine, Marcus Institute of Integrative Health at FAU Medicine, Florida Atlantic University, Boca Raton, Florida, United States of America, 5 Independent Patient Partner, Panama, Florida, United States of America, 6 Independent Family Partner, Panama, Florida, United States of America

* lillian.hung@ubc.ca

**Data Availability Statement:** All relevant data are within the manuscript and its supporting information files.

## Abstract

Older people living with dementia or mild cognitive impairment (MCI) are more vulnerable to experiencing social isolation and loneliness due to their cognitive and physical impairments. Increasingly integrating technology into group exercises contributed to the improved resilience and well-being of older adults living with dementia and MCI. The purpose of this scoping review was to identify the various types, feasibility, outcome measures, and impacts of technology-based group exercise interventions for people with dementia or MCI. We utilized the Joanna Briggs Institute approach, a three-step process. A comprehensive literature search on five databases—CINAHL, MEDLINE, Embase, Web of Science, and PsycInfo—until January 2024 yielded 1,585 publications; the final review included 14 publications that recruited a total of 379 participants, with mean age of 69 (SD = 4.21) years to 87.07 (SD = 3.92) years. Analysis of data showed three types of technology-based group exercise interventions for people with dementia or MCI: (a) exergames, (b) virtual cycling or kayak paddling, and (c) video-conferencing platforms. In addition, we identified three key impacts: (a) feasibility and accessibility; (b) physical, psychosocial, and cognitive benefits; and (c) adaptations necessary for persons with dementia or MCI. Our study suggests that technology-based group exercise interventions are feasible and acceptable to persons with dementia or MCI. Future studies should involve individuals with dementia and their caregivers in the design and implementation of technology-based group exercise programs.

**Funding:** This work received support from the following grants awarded to JP: Grant number 2485 from the Marcus Institute of Integrative Health at FAU Medicine (https://www.faumedicine.org/integrative-health/), Grant 1R01AG08925-01 from the National Institute on Aging (https://www.nia.nih.gov/), and Grant 5R01NR019051-04 from the National Institute of Nursing Research (https://www.ninr.nih.gov/). The sponsors did not play any role in the study design, data collection and analysis, decision to publish, or preparation of the manuscript.

**Competing interests:** The authors have declared that no competing interests exist.

## Introduction

Dementia is a degenerative syndrome that characterizes cognitive and functional capacity significant to perform activities of daily life (ADL) [1]. Exercise often provides some physical and psychosocial benefits, such as reducing loss of physical function and reducing pain and improving function and depressive mood, reducing agitation in older people with dementia [2] or MCI [3]. Although people with dementia may face unique challenges (e.g., changes in attention span, judgment, memory, communication), research has shown that they can successfully engage in regular exercise programs with appropriate support. For example, a study [4] showed that participation in exercise by people with dementia mediated the relationship between cognitive impairment and rehabilitative outcomes. People with cognitive impairment enjoy exercise and believe that it is beneficial for their health and cognition [5]. Exercise in this scoping review paper was defined as physical activity as any planned, structured, and repetitive bodily movement produced by skeletal muscle contraction [6, 7]. Group exercise in particular has been shown to have benefits for people with dementia. They tend to prefer group exercise [5], as it fulfils an important need for "building relationships" [8]. Further, group exercise has been seen as a helpful coping strategy, an effort to "keep the focus off the dementia" by having fun through physical activities with others [9].

Unfortunately, the COVID-19 pandemic caused disruption to community-based exercise programs across the world [10]. Various restrictions were put in place, including closure of important services and activities (e.g., community exercise and wellness programs). These restrictions negatively affected people with dementia and their family caregivers [11]. Even as many facets of society begin to open and lift restrictions, people with dementia and their caregivers continue to exercise precaution and are hesitant to meet in groups in person due to vulnerability to COVID-19 and associated complications [12]. People with dementia are more likely than people without cognitive impairment to experience social isolation and loneliness [13]. Social isolation has been associated with increased risk of depression, cardiovascular complications, and overall adverse effects to quality of life [14].

Technology can serve as a pivotal tool in enhancing access to healthcare for individuals with dementia living in the community, by reducing travel expenses and saving on transportation time for health professional visits [15]. Additionally, it offers innovative alternative exercise options, such as online yoga sessions and virtual reality exercise games, catering to the unique needs of this population [16, 17]. The usage of technology by older people has increased rapidly in recent years [18] During the COVID-19 pandemic and beyond, technology has become essential to maintaining the health of persons living with dementia. Telemedicine by video conference during the COVID-19 pandemic was associated with improved resilience and well-being in people with cognitive impairment [19]. Technology-based group exercise programs, or group exercise programs that utilize technology, including chair yoga (CY), benefit the physical well-being of people with dementia by improving mobility and function and support socialization through social engagement and networking, while simultaneously reducing social isolation and loneliness [16].

To date, no systematic review, meta-analysis or scoping review has been conducted on technology-based group interventions that include virtual reality, exergames, or other Internet-based interventions (synchronous or asynchronous) to support people with dementia. Little is known about whether technology-based exercise interventions provide persons with dementia effective means to partake in group exercise. If so, what types of technology-based exercise programs were offered, what worked, how and why, and what outcome measures and lessons were learned? This scoping review addresses this gap by focusing on various types of technology-based interventions across dementia and MCI and their effects. The format of a scoping

review is often used for burgeoning fields with only small amounts of research published [20]. Scoping reviews attempt to map out volume, nature, and characteristics of a topic of research and do not assess quality of the evidence [20]. Since this field is relatively new, with little published research, a scoping review was considered an appropriate framework.

### Review questions

1. What type of technology-based group exercise interventions are available for people with dementia or MCI?

2. Which outcome measures are utilized in existing studies on this topic?

3. How feasible are these technology-based exercise interventions and what are the results?

## Methods

The protocol for this study can be found at https://doi.org/10.1136/bmjopen-2021-055990. The scoping review follows the PRISMA-ScR (Preferred Reporting Items for Systematic Reviews and Meta-Analysis Extension for Scoping Reviews) checklist as shown in the S1 Table [21].

### Study population

We included studies that focused on all types of dementia (e.g., Alzheimer's disease [AD], vascular dementia, Lewy body dementia, Parkinson's disease dementia, frontotemporal dementia) or MCI in people of all ages.

### Concept

This review focused on technology-based group exercise interventions for people with dementia or MCI. We included studies that involved at least one group of people with dementia or MCI. We defined group as any collection of two or more people, including participant-caregiver dyads. Technology-based exercise was considered to be any form of exercise in which technology is integral to the execution of the activity, such as interventions through video-sharing or online platforms, exergaming, or virtual reality exercises. Exercise intervention was defined as a type of physical activity that is structured and repetitive over the period of time, with aiming to improve symptom and health [7]. Any studies that focused solely on cognitive-based interventions were excluded. Although MCI is considered to be distinct from dementia, due to the relative difficulty of studying people with higher levels of cognitive impairment, as well as the fact that MCI is often considered a precursor to dementia, studies that focused on technology-based group exercise interventions for people with MCI were included, as implications learned from MCI-focused studies could be relevant for populations with dementia [22].

### Context

We included studies in which the intervention took place in homes, assisted living facilities (ALFs), or memory care facilities. We did not include studies in which interventions took place in a hospital or research facility because we wanted to focus on interventions that took place where people lived, and were therefore most comfortable and familiar with their environment.

### Research team for scoping review

The research team consisted of academic scholars and student trainees who work with people with dementia and their caregivers. Participants and family partners were recruited from memory disorder clinics. During the research meetings, we discussed and reflected, utilizing the "ASK ME" ethical framework, which was specifically developed to facilitate co-research with people with dementia [23]. The input by participants and family partners deepened the researchers' understanding of the topic and highlighted the importance of lived experience perspectives. Two of the authors independently performed an initial study selection by assessing the titles and abstracts of identified studies, a third author resolved any conflicts. The authors examined full text of all articles that met the inclusion criteria and completed a summary of characteristics of the studies. The entire research team, including participants and family partners, took part in discussing and analyzing the data from the selected studies during research meetings.

### Search strategy

We followed a three-step search strategy, as recommended by JBI methodology [24]. The search strategy is designed to locate both published and unpublished studies. A comprehensive literature search was conducted between July 2021 and January 2024 to identify relevant peer-reviewed articles using the following electronic databases and keywords. An initial limited search of MEDLINE, CINAHL, and Embase used these selected keywords: (dementia OR (Alzheimer disease) OR (mild cognitive impairment)) AND ((web-based intervention) OR (exergaming) OR (exergame) OR (virtual reality) OR (technology-based intervention) OR telehealth OR telemedicine) AND (exercise OR (physical activity)). Besides, a Cochrane Library search using the keywords above found no relevant systematic reviews available on this topic.In the second step, the following electronic databases were searched for all identified keywords contained in the titles and abstracts of relevant articles and the index terms used to describe the articles: CINAHL (via EBSCOhost, 1982 to 25/January/2024); MEDLINE (via Ovid, 1946 to 25/January/2024); Embase (via Ovid, 1974 to 25/January/2024); Conference Proceedings Citation Index–Science (via Web of Science, 1990 to 25/January/2024); Conference Proceedings Citation Index–Social Science & Humanities (via Web of Science, 1990 to 25/January/2024); Emerging Sources Citation Index (via Web of Science, 2015 to 25/January/2024); PsycInfo (via EBSCO, 1957 to 25/January/2024). The search strategies applied keywords and subject headings to harvest relevant literature on technology-based group exercise interventions for people living with dementia or MCI. The following websites were also searched: AgeWell (https://agewell-nce.ca/ 25/January/2024). There were no articles for which it was necessary to contact the original authors for identification. Finally, the reference lists of all included articles were screened for additional relevant studies. An example of one full electronic search strategy can be found in S1 Appendix.

### Study selection and reviewing results

Identified articles were collated and uploaded into Mendeley, a bibliographic reference management tool, to ensure that references and articles were systematically managed. Once all identified articles were uploaded, duplicate articles were removed. The review process was a two-step process. First, two researchers independently performed a title and abstract review, screening for relevancy; third reviewer resolved any conflicts. Second, relevant articles underwent full text review with examination against inclusion criteria: (a) focusing on people with dementia or MCI, (b) setting at home or in ALFs, (c) included group physical exercise, and (d) used a form of technology to implement exercise. We included studies published in English,

with no time limit on publication date. A range of study designs was included, including qualitative and quantitative designs, randomized controlled trials (RCT) and descriptive studies. Articles were excluded if they (a) included individual physical exercise, not group exercise, (b) were not intervention trials, (c) were not reported in English, or (d) consisted of a single case report or study protocol that did not report results.

The initial database search yielded 1,585 publications; no additional publications were identified through Google search. After removal of duplicates (n = 435), 1,150 articles were screened by title and abstract and 73 articles were identified as potentially relevant. At full text screening, 59 articles were excluded further due to the following reasons: wrong setting (hospital or research facilities; n = 2), Wrong intervention (no exercise or innovative technology involved; n = 17), wrong study design (case report or protocol; n = 2), exercise not group-based (n = 25), wrong participants (older people without dementia or MCI; n = 13). After discussion of eligibility of the articles with participants and family partners, the final review included 14 publications. See Fig 1 for the PRISMA flow diagram [25] that details the review process.

## Mapping

We mapped the selected articles into a table that summarized the articles by author(s), publication year, country, aim, population, sample size, setting, research design, intervention type, outcome measures, and feasibility and results.

## Data synthesis

We adopted the narrative synthesis method to accommodate studies with different designs and reporting styles. The final set of results were analyzed and synthesized by research design, population, sample size, intervention type, outcome measures, and impact and results. We included narrative synthesis that adopts a textual approach and provides both a summary of the knowledge-base and rigorous evaluation, providing a robust interpretative synthesis of the efficacy of the intervention in question [26]. The extracted data were assessed and categorized to create themes that were corroborated by participants and family partners. The data were extracted by two researchers independently, followed by collaborative discussions and synthesis involving the entire research group. This approach ensured a rigorous and comprehensive analysis of the data, as multiple perspectives were considered and synthesized collectively to enhance the validity and reliability of the findings.

## Ethical considerations

This scoping review did not require research ethics approval or consent to participate because the methodology consisted of analyzing data from articles in the public domain.

## Findings

Table 1 outlines the characteristics of 14 eligible studies, with eight originating from the United States [16, 27–33], two from Australia [17, 34], and one each from Germany [35], Korea [36], Taiwan [37], and Brazil [38]. Among these, nine were quantitative studies (including three RCTs [31, 36, 37], five pre-post intervention studies[16, 27, 28, 33, 34], and one cross-sectional design [38]), three were qualitative studies [29, 32, 35], and two were mixed-method studies, one of which was a pre-post intervention design [30], and the other a randomized controlled crossover design [17]. Seven studies were pilot studies [16, 17, 27–30, 33]. One co-design study involved three stages: a pre-study, technology design, and investigation of interaction with the

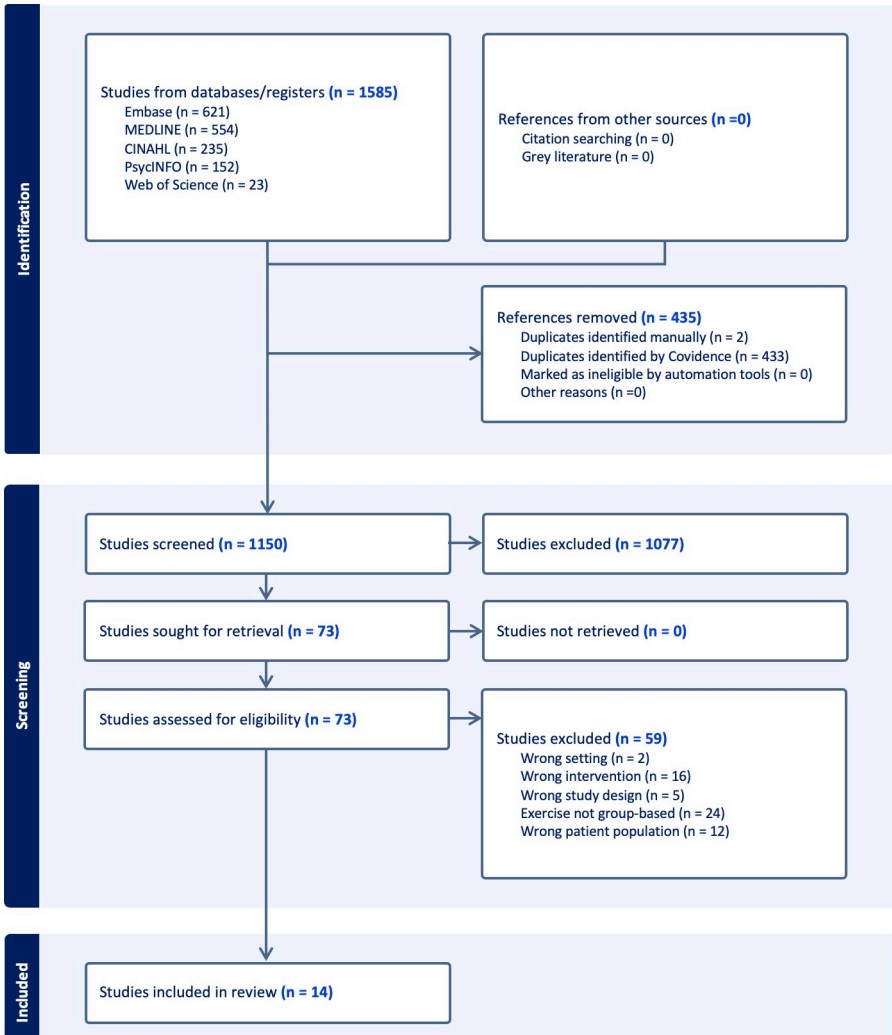

**Fig 1. PRISMA flow diagram for the scoping review.**

technical artifact [35]. Six studies were identified by researchers as feasibility trials [16, 17, 27, 31, 33, 34]. Analysis of these studies yielded themes including design, population and setting, interventions, outcomes measured, feasibility and results.

## Design

The majority of the studies included in the scoping review were primarily aimed at assessing feasibility and gathering preliminary data. Consequently, more quantitative or mixed studies were too small to include a control group. Specifically, six studies utilized a quasi-experimental pre-post intervention one-group design [16, 27, 28, 30, 33, 34]. Studies that had control groups often used a type of exercise that did not involve technology as the control; this allowed them to assess the effect of the technology. Of the studies with control groups, traditional Tai Chi (TC) [37], stretching [31], a seated group mobility exercise[17], home exercise [36] and no exercise [38] were provided as controls. Four trials utilized an RCT design. One was a mixed-methods randomized controlled, crossover feasibility study design, randomized to either virtual cycling or a control group of seated group mobility exercise sessions [17]. ALL three other

**Table 1. Charting table of scoping review.**

| Author, Year, Country | Aim | Population | Research Design | Intervention | Outcome Measures | Feasibility and Results |
|---|---|---|---|---|---|---|
| Jhaveri S (2023) USA [30] | To explore the impact of SMARTfit exergaming system in physical and cognitive functions of older adults with MCI | Older adults with MCI living at home (N = 18) | Mixed design (Pre-post intervention study) | SMARTfit dual-task training (DTT; 12 games with various focus areas) 60 min per week for 12 weeks | • Experience assessment (10-item survey) • Cognitive function (TMT and SCWT) • Physical function (SPPB) | The SMARTfit exergaming system enhances executive function, interference inhibition, and physical function in older adults with MCI, consequently benefiting their quality of life. Participants reported finding SMARTfit enjoyable, interactively easy to use, and challenging. The preferred game duration was 2 minutes. They expressed a preference for the fast-paced and unpredictable nature of certain games over others, with "memory" being less favored. |
| Park J (2023) USA [32] | To identify benefits, challenges, and facilitators for older adults with dementia participating online chair yoga (CY) exercises | Older adults with dementia and caregivers living at home (N = 17; 8 people with dementia and 9 family caregivers) | Qualitative study | Remotely supervised CY session, twice a week for 8 weeks | Focus groups (study design at pre-intervention; challenges faced by participants at post-intervention) | Online yoga exercises enhance sleep quality, physical and mental health, and social connections among older adults with dementia, despite technological setup challenges. Additionally, online chair yoga proves feasible and acceptable for socially isolated older adults with dementia, offering convenience and relaxation in their home environment. |
| Sari YM (2023) Indonesia [34] | To evaluate the feasibility of the telehealth exercise program for older people with dementia | Older people with dementia and their carers living at home (N = 60; 30 residents, 30 caregivers) | Pre-post intervention study | 20-30-min supervised telehealth exercise program, 5 days a week for 12 weeks (modified Otago exercise program for prevention of falls) | • Physical activity level (PASE) • Function and disability (LLFDI) • Health-related benefits (Vitality Plus Scale) • Fear of falls (Icon-FES) • Enjoyment (PACES) • Quality of life (QOL-AD) • Neuropsychiatric symptoms (NPI-Q) • Impact on carers (Zarit caregiver burden scale) | Five efficacy outcomes, including physical activity level, function and disability, health-related benefits, enjoyment, and quality of life, demonstrate significant enhancement through the telehealth exercise program for individuals with dementia. Moreover, the program's feasibility and safety for this demographic are underscored, highlighting its potential as a reliable intervention |

(*Continued*)

**Table 1.** (Continued)

| Author, Year, Country | Aim | Population | Research Design | Intervention | Outcome Measures | Feasibility and Results |
|---|---|---|---|---|---|---|
| Liu CL (2022) Taiwan [37] | To explore effects of exergaming TC training on physical and cognitive in older adults with MCI. | 50 participants with MCI from communities | RCT | 36 exergaming based Tai Chi sessions (50-min sessions, thrice a week for 12 weeks); control groups take traditional TC or no exercises. | • Cognitive function (MoCA, TMT, CCVLT, SCWT, and One-back test)<br>• Gait performance and dual-task costs (wearable GAIT UP data) | Exergaming group had improved cognitive performance and better gait speed and cadence than control group. Exergaming facilitates the positive effects of TC and shows potential therapeutic benefits in older adults with MCI. |
| Nora CD (2022) Brazil [38] | To explore the effect of online physical exercise program on neuropsychiatric symptoms and quality of life of patients with dementia | Older adults (N = 25; 14 AD, 8 other dementia, 2 MCI; 14 in online exercise group and 11 in control group) with their caregivers | cross-sectional study | 60-min remotely supervised physical exercise (via Zoom platform) in groups of a maximum of 10 participants twice a week for more than 3 months while the control group did not take any exercise. | • Neuropsychiatric symptoms (NPI-Q)<br>• Quality of life (QOL-AD scale) | Participants engaged in an online physical exercise program during the pandemic experienced fewer neuropsychiatric symptoms, reduced nighttime disturbances, and reported enhanced subjective memory. This intervention demonstrates promise as a feasible approach for mitigating sedentary behavior and ameliorating behavioral symptoms in older people with dementia or MCI. |
| Park J (2022) USA [16] | To evaluate the feasibility of remote online Chair Yoga intervention for older adults with dementia and impact on relevant clinical outcomes | Older adults living at home with dementia (N = 11) | pre-post intervention study | Remotely supervised chair yoga 60-min session, twice a week for 8 weeks | • Feasibility (i.e. retention, adherence, safety)<br>• Clinical outcomes (PROMIS PI-SF, SDI, de Jong Gierveld Loneliness Scale, BPQ-SF, TUG)<br>• Cardiac data | remote online CY is feasible with retention (70%) and adherence (87.5%). No changes on pain interference, mobility, sleep, and social loneliness has been witnessed while emotional loneliness increased during pandemic period.<br>A twice-weekly CY session for 8 weeks may not be sufficient for older adults with dementia to decrease isolation. Telehealth-based CY intervention was found to be convenient to both participants and their caregivers. |

**Table 1.** (Continued)

| Author, Year, Country | Aim | Population | Research Design | Intervention | Outcome Measures | Feasibility and Results |
|---|---|---|---|---|---|---|
| D'Cunha NM (2021) Australia [17] | To evaluate safety and feasibility of virtual cycling experience (VCE) and benefits on mood, apathy, and engagement, and to explore perceptions of and facilitators to people with MCI | Older adults with MCI living in residential aged care facilities (N = 10, 5 in VCE-control condition group, 5 in control condition-VCE group) | Mixed study design (randomised, controlled, crossover study) | VCE intervention takes 25-min virtual cycling exercise while control condition takes 25 min seated group mobility exercise | • Pre- and post-intervention mood questionnaire<br>• Person-Environmental Apathy Rating scale<br>• Engagement of a Person with Dementia scale<br>• virtual cycling experience (Semi-structured interviews) | Environmental stimulation was the only aspect found to be lower in the Virtual Cycling Environment (VCE) compared to the control condition, with a lower response observed in the intervention group. Participants described VCE as immersive and challenging, often recalling earlier cycling experiences. The activity manager emphasized the benefits of safety screening and thorough preparation before engaging in the activities. |
| Li F (2021) USA [31] | To explore the feasibility of an online balance training exercise program on older adults with MCI | Older adults with MCI at home (N = 30; 15 in Tai Chi group, 15 in control group) | RCT | 60-min virtual exercise sessions (via Zoom) twice weekly for 24 weeks while control group takes stretching exercise | • Incidence of falls<br>• Balance (4-stage balance test, 30-second chair stands, TUG) under both single- and dual task conditions | The Tai Chi group did not reduce the incidence of falls but demonstrated improved balance among older adults with MCI, with a commendable attendance rate of 79%, indicating good feasibility. |
| Unbehaun D (2021) Germany [35] | To investigate how music-based exergames affect quality of life in people living with dementia | Older adults with dementia and their caregivers in day-care facilities and private households (N = 20; 10 residents living with dementia, 10 caregivers) | qualitative study (Co-design workshops) | Interactive music exergame twice a week for 4 months (dancing, walking, e.g.) | • 12 semi-structured 60-min interviews were conducted focusing on 1) perception of the exergame and experience of interaction; 2) changes and impact by exergame system, like individual mood, well-being, social dynamics, activity levels, and stimulated communications. | The music-based exergame foster memories and social interaction and improve physical activity in people living with dementia and their caregivers |
| Choi W (2019) Korea [36] | To investigate the effects of a virtual kayak paddling (VKP) exercises on postural control, muscle performance and cognitive function in older adults with MCI | Older adults with MCI living at communities (N = 60; 30 in VKP; 30 in control group) | RCT | VKP exercise 60 min/day, twice a week for 6 weeks; control group taking home exercise | • Static balance (one-leg stance test)<br>• Dynamic balance (TUG, FRT, BBS, FSST)<br>• Muscle performance (Arm curl test)<br>• Cognitive function (MoCA and GPCOG) | Postural balance, muscle performance, and cognitive function were improved in VKP group. VKP exercise improved balance, strength and cognitive function in older adults with MCI. |
| Ptomey LT (2019) USA [33] | To evaluate the feasibility of a group video conference approach for increasing moderate-intensity physical activity (MPA) in adults with AD and their caregivers | Older adults with AD and their caregiver dyads at home (N = 18) | pre-post intervention study | 30-min remotely delivered group exercise sessions three times a week for 12 weeks | • MPA time (Fitbit Charge HR monitor)<br>• Quality of life (QOL-AD)<br>• Survey (post intervention) | Exercise delivered by group video conferencing is feasible and effective for increasing MPA in adults with AD. |

(*Continued*)

**Table 1.** (Continued)

| Author, Year, Country | Aim | Population | Research Design | Intervention | Outcome Measures | Feasibility and Results |
|---|---|---|---|---|---|---|
| Anderson-Hanley C (2018) USA [27] | To examine neuropsychological and neurobiological outcomes of interactive physical and mental exercise | Older adults with MCI living at home or senior living facilities and their co-residing pairs (N = 31) | pre/post-test study | 20-40-min iPACES exergame was conducted 3-5x/week for 3 months after 2 week familiarization period at the beginning. | • Executive function (Stroop A/C, Digit Span B/F, and Color Trails 1/2) • Verbal memory (ADAS) • Cognitive function biomarker (Saliva cortisol and IGF-1) | The iPACES exergame is feasible for community-dwelling residents with MCI and improves their executive function and verbal memory if full dose (> = 2 per week) can be fulfilled. No changes on cognitive biomarkers were observed. |
| Wall K (2018) USA [28] | To explore whether an intertwined physical and mental exercise could improve cognitive performance (executive function) in older adults with MCI. | Older adults with MCI living at home (N = 14; 7 pairs) | pre/post-test study (Pilot) | 2-week placebo, 2-week physical exercise, 2-week cognitive exercise, and 7-week iPACES exergame (totally 3 months) continuously, 30–45 min/week for 3–5 times per week. | • Executive function (BrainBaseline Stroop, trails, and flanker test) • Verbal memory (ADAS) • Global cognitive function (MoCA) • Cognitive function biomarker (Saliva cortisol and IGF-1) | The iPACES exergame could improve executive function (Stroop test) in older adults with MCI. Biomarkers were associated with improved cognition. |
| Chao YY (2016) USA [29] | To explore the facilitators and barriers to applying exergames exercise on assisted living residents | 15 assisted living residents (N = 15; 4 of them living with dementia; 4 males and 11 females) | qualitative study | 60-min Wii Exergames exercise were conducted to participants in teams of two, twice a week for 4 weeks (jogging, lunge, penguin slide, e.g.) | • Semi structural individual interview responses were analyzed with basic content analysis methods. Content included motivation, barriers, participation reasons, cognitive, physical, and psychological effects of Wii Fit exergames. | Facilitators (being healthy and alert, positive mindset, social interaction, and well-structured program) and barriers (physical impairment and unpleasant past experiences) to exercise were addressed. Participants identified positive physical, cognitive, and psychological benefits of exergames. |

Note: MoCA = Montreal Cognitive Assessment, GPCOG = General Practitioner Assessment of Cognition; TMT = Trail Making Test, CCVLT = Chinese version of California Verbal Learning Test; SCWT = Stroop Color and Word Test, SPPB = Short Physical Performance Battery; OLS = one-leg stance; TUG = timed up and go test; FRT = functional reach test; BBS = Berg Balance Scale; FSST = Four Square Step Test, PASE = Physical Activity Scale for the Elderly; LLFDI = Late-Life Function and Disability Instrument; Icon-FES = Iconographical Falls Efficacy Scale; 8-item PACES = 8-item Physical Activity Enjoyment Scale; QOL-AD = Quality of Life in Alzheimer's Disease; NPI-Q = Neuropsychiatric Inventory Questionnaire; PROMIS PI-SF = PROMIS Pain Interference-Short Form, SDI = Sleep Disorders Inventory, BPQ-SF = Body Perception Questionnaire-Short Form, TUG = Timed Up and Go, ADAS = Alzheimer's Disease Assessment Scale, iPACES = Interactive Physical and Cognitive Exercise System.

RCT studies were single-blinded, randomized, parallel-group trials [31, 36, 37]. In total, six of the studies were researcher-identified as feasibility trials, including a feasibility pilot study in which participants were assigned to exercise sessions via video conferencing in addition to one education session with a health coach [33]. One case study involved three phases: (a) a pre-study that consisted of an analysis of existing practices in the field, (b) a design of the technological artifact based on findings from Phase 1, and (c) an investigation of interaction with the artifact over time [35].

## Population, sample size, setting

Overall, the studies had small sample sizes, which is consistent with the fact that most were feasibility studies. The 14 selected studies recruited a total sample of 379 participants (Table 2), with mean age ranging from 69 (SD = 4.21) [30] to 87.07 (SD = 3.92) [29]. Study sample sizes ranged from 10 [17] to 60 [34]. In terms of the ratio of sex, 4 of the 14 studies had fewer women than men in terms of residents living with dementia or MCI [16, 28, 32, 38]. Several studies did not report race/ethnicity of participants [17, 34–38]. Studies that did report ethnicity/race generally had a majority of White/Caucasian participants [16, 27–29, 31–33]. Seven studies focused on participants with MCI [17, 27, 28, 30, 31, 36, 37]. One study focused on ALF residents and included persons with dementia [29]. One study focused on older adults with cognitive impairment but did not require a formal dementia diagnosis, utilizing a Montreal Cognitive Assessment (MoCA) score to screen for cognitive impairment [16]. Three studies broadly included all types of dementia [32, 34, 38]; participants in the other study included persons with AD and their caregivers [33]. In terms of setting, the majority of the studies took place in domestic settings, such as private households, ALFs, or senior living facilities. Several studies took place solely in homes [16, 28, 30–34, 36–38], while other studies took place in homes and senior living facilities [27] or homes and daycare centers [35]. Several studies took place only in senior living facilities, specifically in ALFs [29] and residential care facilities [17].

**Table 2. Demographic characteristics of participants.**

| Author, Year | Sample Size | Age | Gender (N[%]female) | Ethnicity | Cognitive Level | Dementia Types |
|---|---|---|---|---|---|---|
| Jhaveri S (2023)[30] | Phase 1: 8<br>Phase 2: 10 | 69.0 ± 4.21<br>73 ± 4.64 | 6 (75%)<br>7 (70%) | African American: 18 | MCI | |
| Park J (2023) [32] | 17 (PwD 8) | 80.73 ± 8.7 | 3 (37.5%) | Non-Hispanic White: 7<br>African American: 1 | dementia | Alzheimer's disease: 2<br>Lewy body dementia: 1<br>Other 5 |
| Sari YM (2023) [34] | 60 (PwD 30) | 71.5 ± 8.8 | 19 (63.3%) | Not reported | dementia | Alzheimer's Disease: 22<br>Vascular dementia: 6<br>Frontotemporal: 1<br>Parkinson's dementia: 1 |
| Liu CL (2022) [37] | 50 | 74.6± 6.1 | 35 (70%) | Not reported | MCI | |
| Nora CD (2022) [38] | 25 | 78.0 ± 4.0 | 12 (48%) | Not reported | Dementia and MCI | Alzheimer's Disease: 14<br>Vascular disease: 1<br>Lewy bodies: 1<br>Frontotemporal: 1<br>Unspecified: 5 |
| Park J (2022) [16] | 11 | 80.8 ± 7.458 | 3 (27.3%) | Caucasian: 10<br>African American: 1 | dementia | Not reported |
| D'Cunha NM (2021) [17] | 10 | 86.1 ± 8.06 | 8 (80%) | Not reported | MCI | |
| Li F (2021) [31] | 30 | 76.13 ± 6.2 | 21 (70%) | White: 27<br>Others: 3 | MCI | |
| Unbehaun D (2021) [35] | 20 (PwD 10) | 83.3 ± 5.68 | 8 (80%) | Not reported | dementia | Not reported |
| Choi W (2019) [33, 36] | 60 | 77.27 ± 4.37 | 51 (85.0%) | Not reported | MCI | |
| Ptomey LT (2019) [33] | 18 (PwD 9) | 74.1 ± 10.2 | 5 (55.6%) | Black: 1<br>White: 8 | dementia | Alzheimer's Disease: 9 |
| Anderson-Hanley C (2018) [27] | 31 | 76.1 ± 10.4 | 18 (58.1%) | Caucasian: 31 | MCI | |
| Wall K (2018) [28] | 14 | 82.8 ± 3.9 | 6 (42.9%) | Caucasian: 14 | MCI | |
| Chao YY (2016) [29] | 15 | 87.07 ± 3.92 | 11 (73.33%) | Caucasian: 15 | Dementia | Not reported |

Note: PwD = Participants with Dementia; MCI = Mild Cognitive Impairment.

## Types of interventions

**Virtual cycling or kayak paddling.** A common form of exercise is cycling; cycling is familiar to many people and is easily translated into a virtual activity via an appropriate setup with a screen and stationary bicycle. Virtual cycling was identified as a technology-based intervention in three articles [17, 27, 28]. Virtual cycling entails pedalling on a fixed seat while watching a screen that shows a corresponding virtual bike path and scenery. All virtual cycling was used with people with MCI. Two studies specifically utilized the Interactive Physical and Cognitive Exercise System (iPACES) intervention [27, 28]. This intervention involves pedalling and steering along a virtual bike path to complete an element of running errands. The virtual cycling interventions were installed either at home [27, 28] or in residential facilities [17]. In one study, participants with dementia were instructed to cycle for one 25-minute session [17]; participants with MCI in the iPACES trials were told to practice iPACES for 20–40 minutes 3–5 times per week for a 2-week familiarization window, and then continue for 3 months [27, 28]. Virtual kayak paddling, akin to virtual cycling, utilizes similar equipment and engagement strategies, employing a soft balance foam on a chair and a 1-kg paddle [36]. Two studies utilized filmed videos of lakes or mountains and 100-inch projectors positioned at a 3-meter distance to enhance the virtual reality effect [17, 36].

**Video conferencing platform.** Video conferencing platforms are a relatively new way to deliver long-distance group exercise programs [16, 31–34, 38]. The video conferencing platform Zoom was employed to deliver virtual real-time group exercise synchronously to persons with AD and their caregivers by a trained health coach for three 30-minute sessions per week for 12 weeks [33]. Participants were given iPad mini tablet computers (Apple Inc., Cupertino, CA) with the video conferencing software Zoom downloaded. Zoom was also used to deliver synchronous augmented dual-task TC or a stretching program to older adults with MCI for two 1-hour sessions per week for 24 weeks [31]. Zoom was used to deliver synchronous CY to persons with dementia and their caregivers for two 1-hour sessions per week for a total of 8 weeks [16, 32]. Sari et al. [34] facilitated home exercise sessions (resistance, balance, walking) by setting up Zoom software, camera, and sound equipment and provided training and supervision to participants in four out of the total 12 sessions. Finally, Zoom was used to deliver aerobic exercise to persons with dementia and their caregivers twice a week for 60 minutes a week for at least 3 months [38]. All of the video conferencing platform interventions were used in home settings [16, 31–34, 38].

**Exergames.** Exergames are video games that rely on various types of physical exercise; they often use gaming consoles as a medium to exercise. Exergames were used as technology-based exercise interventions in four studies [29, 30, 35, 37]. Several of the exergames were implemented through the Wii video game console and corresponding Wii balance board, as participants watched the Wii screen while standing on the balance board to exercise [29]. The Wii interventions involved 1-hour sessions twice a week for 4 weeks. The music exergame had fixed sessions one or two times per week, but participants could play more or less often, depending on their mood and health [35]. Liu et al [37] leveraged the infrared light component of the Kinect system to create a virtual full-body 3D map and participants replicated the movements of a virtual TC coach and made real-time adjustments to their movements in response to immediate feedback. Jhaveri et al. [30] utilized a combination of 12 training games with various focus areas to engage participants with dual-task training (DTT). Exergames were used in ALFs [29], in homes [30], in communities [37] and daycare facilities [35]. Two studies involving exergames included people with dementia [29, 35].

## Outcome measures

Nine quantitative studies focus on comparing cognitive or physical function variables alongside psychological and behavioural variables, while three qualitative studies explore user experiences, highlighting challenges and facilitators in technology-based group exercises. Additionally, two mixed-methods studies integrate both quantitative and qualitative approaches to provide a comprehensive understanding of the subject matter. As a result of varying intervention types, outcome measures and results will be categorized and summarized according to distinct variables corresponding to three different types of interventions.

**Cognitive outcomes.** Several studies investigated the potential link between technology-based exercise interventions and cognitive improvement. Given that areas of cognition such as attention, memory, and executive function are often negatively affected in dementia, these aspects were the focus of research attention. Cognitive function was assessed in five studies involving participants with MCI [27, 28, 30, 36, 37]. Notably, no studies involving individuals with dementia reported cognitive function outcomes. Various outcome measures evaluating different aspects of cognition were utilized. Notably, cognitive measures were absent in studies examining interventions using video conference platforms. Two studies employing exergames reported improved cognitive function in older adults with MCI. Specifically, exergames with DTT led to improvements in executive function (Trail Making Test, TMT) and interference avoidance (Stroop Color and Word Test, SCWT) [30], while exergames based on TC exercises improved global cognition (MoCA), attention (SCWT), executive function (TMT), and working memory (one-back test) overall [37]. Two studies conducted by the same researchers on virtual cycling (iPACES system) reported cognitive improvements in executive function (Stroop A/C ratio) and verbal memory (Alzheimer's Disease Assessment Scale (ADAS)-delayed recall), particularly with a full dosage ($\geq$ 2 sessions per week, 20–40 minutes per session) and larger sample sizes [27], whereas these differences were not observed in a pilot study with a smaller sample size [28]. Choi et al. [36] reported improved cognitive function, as measured by MoCA and General Practitioner Assessment of Cognition (GPCOG) variables, when virtual kayak paddling was introduced to older adults with MCI.

**Physical outcomes.** Since dementia can often cause diminished physical function and has been associated with an increased risk of falls [39], five studies attempted to assess the impact of technological interventions on physical function in persons with dementia or MCI. In the exergame intervention, Jhaveri et al. [30] used the Short Physical Performance Battery (SPPB) to assess the balance, strength, and gait speed of lower extremity function, while Liu et al. [37] used the wearable GAIT Up system to assess gait performance under three circumstances (single task, cognitive dual-task, motor-dual task). Both studies showed positive results on physical function for older adults with MCI. While applying exergames to TC improved lower extremity strength and balance [37], exergames with Dual-Task Training (DDT) in group exercise could improve gait speed and cadence for older adults with MCI [30]. For the virtual kayak paddling intervention, strength (Arm Curl Test), static balance (One-Leg Stance Test), and dynamic balance (TUG, FRT, BBS, FSST) were assessed in older adults with MCI, and positive results with better postural balance and muscle performance were reported [36]. In the online video conference platform intervention, two studies applying instructor supervision to TC and CY reported positive physical outcomes [16, 31]. Li et al. [31] assessed the number of falls and other balance outcomes (4-Stage Balance Test, 30-Second Chair Stands, TUG) under both single and dual-task conditions and reported that TC online group exercise did not reduce the incidence of falls but improved balance among older adults with MCI. One study on online CY exercise is the only study which assessed physical outcomes for older adults with dementia, and the TUG as the outcome of balance assessment showed better balance at online CY group

[16]. In summary, various types of technology contribute to better physical function in older adults with dementia or MCI, and balance is the most common physical function variable evaluated with TUG as the most popular approach to assess balance level.

**Psychological and behavioral outcomes.** People with dementia often experience Behavioral and Psychological Symptoms of Dementia (BPSD), such as depression, anxiety, apathy, agitation, and irritability. Psychological outcomes were measured in six studies [16, 17, 29, 33, 34, 38]. Engagement and apathy were assessed in a study on virtual cycling [17] via the Person-Environment Apathy Rating Scale [40] and the Engagement of a Person with Dementia Scale [41]. Neither of these assessments indicated significant changes. Diminished quality of sleep, increased pain interference in daily life, and loneliness are common symptoms in dementia that can affect physical and mental health [42]. The effects of technology-based exercise on these symptoms were measured in one study [16]; quality of sleep was measured by the Sleep Disorders Inventory [43], interference of pain in daily life was measured by the PROMIS Pain Interference Short Form [44], and loneliness was assessed via the de Jong Gierveld Loneliness Scale [45]; none of these showed significant differences over time. Another study [38] measured neuropsychiatric symptoms of dementia, such as hallucinations, agitation, and agitation, via the Neuropsychiatric Inventory [46]. This study showed significantly fewer psychiatric symptoms overall in those who participated in the remote group exercise intervention, with individual analysis of the scale items showing significant improvement in nighttime behavior disturbances [38].Fear of falling and belief in the capacity to exercise are important psychological metrics. Sari et al. [34] reported significant enhancements in physical activity level (PASE), function and disability (LLFDI), health-related benefits (Vitality Plus Scale), fear of falls (Icon-FES), and enjoyment (PACES), and clarified the feasibility of the online video conference platform for older adults with dementia. Some measures of quality of life were utilized. Overall health has important implications for quality of life; quality of life was assessed via QOL-AD in three studies [33, 34, 38], and improvement was reported to be brought about by the online video conference approach to older adults with dementia or MCI, with consistency among those studies. To summarize, all three interventions contributed to improvements in quality of life, yet variations in psychological and behavioural outcomes were observed across various studies.

**Biomarkers.** Dementia biomarkers are useful for diagnosis and monitoring progression across various forms of dementia. The quest for newer, more precise, sensitive, and minimally invasive biomarkers is a significant focus in dementia research. Salivary biomarker assays were evaluated in two studies, one of which was a pilot study, conducted by one research group employing the iPACES system [27, 28]. While a correlation between biomarkers cortisol and insulin-like growth factor 1 (IGF-1) and altered cognitive function levels was identified, no changes of those biomarkers were observed upon introduction of the iPACES exergame.

## Feasibility and results

Within the scoping review, the feasibility studies showed that technology-based exercise interventions for people with dementia or MCI were feasible and accessible. Measures of feasibility were based on program fidelity, program compliance, and low attrition rate. Five of six feasibility studies achieved retention rates of more than 70%, and none reported serious adverse events [16, 17, 31, 33, 34]. In the iPACES exergame study, 11 out of 31 participants discontinued the intervention due to finding the game either too difficult, too easy, not enough time, or unrelated health issues [27]. Additionally, one study reported that 86.7% of participants completed a 12-week online video conferencing exercise led by a physiotherapist [34]. In another study on TC class participation, the rate was 79%, with a 13% attrition rate [31]. Another

Zoom exercise study [33] achieved a 78% completion rate, with participants with AD attending an average of 77.3% of the group exercise sessions. In a virtual cycling feasibility study, 7 out of 10 participants finished the intervention, with 3 stopping due to lower body discomfort [17]. Lastly, the remote CY feasibility study had a retention rate of 70%, with an adherence rate of 87.5% [16]. This study [16] also reported that a limitation to feasibility was that caregivers can also face physical and cognitive challenges that can make interacting with technology difficult. Li et al. also discussed that caregivers felt that the largest barrier to participation in the study was use of technology, with up to 42.9% reporting some type of difficulty with technology; however, 100% of adults with Alzheimer Disease and 100% of caregivers reported that they enjoyed the remotely delivered exercise program and would do it again [31].

A variety of types of technologies (exergames, video conferencing platforms, virtual cycling) were found to be acceptable to persons with dementia or MCI and their caregivers. Acceptability of the interventions was based on results of qualitative methods that directly asked about acceptability [29] or other thoughts and feelings about the interventions [17, 35], in addition to other measures of acceptability such as attrition, as discussed above [31]. In a study that compared the technology-based intervention with usual activity, the technology-based intervention was found to be an acceptable alternative to engaging in usual activities [17]. The interventions could easily be adapted to homes, senior living facilities, and ALFs [27, 33, 35]. Ease of accessibility of the technology-based approach may allow exercise to be provided to persons with dementia or MCI who do not have access to gyms or community programs [33].

Authors of studies in the scoping review stated that key adaptations should be made in order to adopt technology-based exercise interventions for people with dementia or MCI. A study of virtual cycling emphasized the benefits of safety screening and proper preparation prior to initiation of activities [17]. Similarly, a study that utilized a music exergame intervention indicated that guidance is required for persons with dementia to play the exergame, as they do not always remember how to use the intervention without assistance [35]. In another study of virtual cycling, it was reported that some participants found the exergame to be difficult, while others reported that it was too easy [27]. Thus, the study results indicated that the intervention should be adapted to the level of the individual participant, in view of many stages of dementia and cognitive impairment.

Results from interviews in the qualitative studies indicated that participants identified positive physical, cognitive, and psychosocial benefits of exergames [29, 35]. Participation in a virtual cycling intervention led to increased interaction with people in and out of the facility, increasing social interaction and reducing social isolation [17]. Older adults with MCI participating TC exergame reported potential therapeutic benefits [37]. Persons with dementia or MCI who participated in technology-based exercise interventions reported that the intervention fostered memories and led them to reminisce about previous experiences with cycling, dancing, and singing [17, 35]. Participants also reported that they enjoyed the sense of contribution to research and a sense of choice and empowerment, which promoted self-efficacy through the experience of doing something new [17, 35]. Using technology to exercise in a group at home was rated as convenient by both persons with dementia and their caregivers [16, 32].

## Discussion and implications

This scoping review examined the types of technology-based group exercise interventions applied at home or long-term care facilities. Overall, they were found feasible for people with dementia or MCI. Most studies had small sample sizes, with a majority of participants being White and female. This finding of ethnic/racial disparity is consistent with previous trends in

research [47]. While the higher participation rate of women is in line with the overall higher incidence of dementia in females than males [48], it is important to have racially diverse groups [47, 49]. As small sample sizes are consistent with the purpose of pilot studies [50], larger-scale studies are needed to provide more robust evidence.

The studies included in the review were focused on older adult populations, with mean ages of 69 years to 87.07 years. Given the fact that there are physiological changes associated with aging, such as loss of muscle mass and strength and decreased physical performance [51], research comparing the effects of these technology-based exercise interventions between younger and older adults with dementia should be examined in the future studies. The majority of participants in these interventions resided in their homes in the community. Very few studies were conducted in long-term care facilities. Individuals with dementia residing in long-term care facilities are more likely to have more functional symptoms of dementia compared to those living at home. It is important for future research to investigate strategies to assist these individuals in engaging with technology-based group exercise programs. The impact of workforce shortage in long-term care facilities offers opportunities and limitations for technology-based group exercise programs.

There was little focus in the studies on the effect of the interventions on various types of dementia or the severity of dementia. None of the studies analyzed results by type of dementia. Different types of dementia can have distinct clinical presentations and varied responses to specific therapies due to underlying differential mechanisms. For example, while cholinesterase inhibitors are first-line pharmacologic interventions to treat mild Alzheimer's dementia, Parkinson's disease dementia, and Lewy body dementia, there is differing strength of evidence to support the effectiveness of this medication class for each type of dementia [52, 53]. Likewise, nonpharmacologic interventions such as exercise could have a differential effect depending on the underlying disease process. In people with vascular dementia, in which a core component of treatment is vascular risk factor management, exercise may play a relatively stronger role [54]. Stage and severity of disease have also been shown to influence treatment efficacy; These findings indicate the need for research with a racially and ethnically diverse population, with large sample sizes, and with participants stratified by type of dementia or cognitive impairment.

In the synthesis process, the strategy of identifying key themes and concepts was employed to categorize interventions into three overarching themes. Technology-based group exercises often incorporate "game elements," "immersive experiences," and "remotely supervised" features, all aimed at motivating participants to maintain adherence to exercise. Under this common aim, integrating three key characteristics might potentially maximize engagement effects. Anderson et al. [27] reported that improved executive function and verbal memory can only be observed at a "full dose" (20–40 minutes per session and $\geq$2 sessions per week) instead of an incomplete dose, emphasizing the importance of ensuring exercise adherence by introducing innovative technology into exercise routines to attract and inspire interest. D'Cunha et al. [17] and Choi et al. [36] utilized filmed videos of lakes and mountains and 100-inch projectors to enhance immersive effects of virtual reality, reported improved postural balance, muscle performance and cognitive function, and positive experience on older adults with MCI. One study utilizing Oculus Rift head-mounted display and Oculus touch controllers to create a virtual reality environment reported that VR exergames could be a promising technology for engaging people living with dementia [55]. Another study attempted to develop new VR exergames called "Seas the Day" through multistakeholder co-design, combining exergaming, virtual reality, and online assessment to fully engage persons living with dementia [56]. A systematic review reported that VR exergames provided potentially positive influences on cognition, memory, and depression in older adult populations [57]. Innovative technology,

especially integrating these three types of interventions, needs to be assessed in future research concerning older adults with dementia or MCI. Besides, based on observations from qualitative and mixed-methods studies, participants have articulated their preferences for exercise, highlighting several key characteristics: enjoyable, user-friendly (especially in home environments like CY), appropriately challenging (avoiding monotony), immersive, of suitable duration (around 2 minutes), fast-paced, unpredictable, requiring minimal reliance on memory, socially interactive, and well-structured. These insights serve as valuable guidelines for the development of innovative technology in the field of exercise.

Comparison and contrast strategies were utilized to analyze various resources focusing on outcomes categorized by different functions. This approach involved systematically examining the similarities and differences in the reported outcomes across the diverse resources. By categorizing outcomes based on distinct functions, the synthesis aimed to highlight commonalities and discrepancies among the findings, thereby facilitating a comprehensive understanding of the subject matter. All quantitative studies assessed various cognitive function including memory and executive function, physical function, and psychosocial function (e.g., depression, loneliness), and neuropsychiatric symptoms [17, 31, 36, 37]. Some studies showed significant improvement in outcomes, while others showed nonsignificant results. Significant improvements in psychosocial, cognitive, and physical functions are consistent with findings of previous research on in-person exercise for persons with dementia [58, 59]. Other studies did not find significant changes in certain measures of cognitive function, psychosocial function, or physical function [17, 28, 31]. In order to resolve the inconsistent findings, a rigorous design with a larger sample size is needed. Within the literature included in this review, relatively few studies utilized a blinded RCT design, while many were quasi-experimental or pilot studies. In addition to the paucity of studies, this indicates the relative novelty of the field. Positive results of these pilot studies support the need for further research, with RCT design, to be performed to increase strength of evidence and impact.

The reviewed studies encompassed a spectrum of outcome measures, comprising qualitative psychosocial, physical, and cognitive assessments, alongside biomarkers. Although biomarkers were measured as frequently as other types of outcome markers, the types of biomarkers assessed were not conventional. Salivary biomarkers and heart rate variability were measured, whereas more conventional biomarkers for dementia such as structural brain imaging (CT or MRI), functional imaging (PET) and fluid markers in cerebrospinal fluid were not measured [60]. Even though the biomarkers that were measured were unconventional, one significant advantage was that both saliva and heart rate variability were collected remotely. Measuring biomarkers is crucial for monitoring changes and the progression of dementia. The utilization of remote biomarkers has the potential to eliminate the necessity for individuals with dementia to travel for disease progression monitoring. While both biomarkers are relatively recent, with limited collection and scope, and necessitating further study, the emerging practice of remote biomarker collection enhances accessibility for individuals who might otherwise lack opportunities to participate in research and monitor dementia progression.

However, remote data collection could also have several barriers to implementation. This type of data collection would either rely on researchers traveling to the participant's location, which could be limited by distance, time and funding, or it could utilize caregivers, which could be limited by caregiver ability to learn how to adequately collect data. Further research is needed to fully understand the utility and reliability of these remote biomarkers.

Quantitative outcome measures are crucial for research and medicine, yet qualitative outcomes, including quality of life improvements, are equally if not more vital to individuals with dementia and their caregivers. Using techniques such as interviews, people with dementia

were given the opportunity to discuss how the intervention made them feel and how it impacted their lives. Quality of life was also measured empirically. While QOL-AD can be used to evaluate quality of life on people living with dementia, more specific measures for these populations exist, such as Alzheimer Disease-Related Quality of Life (ADRQL), Quality of Life in Late-Stage Dementia (QUALID), QUALIDEM (a dementia-specific QOL tool), and DEMQOL (health-related QOL for people with dementia) [61]. These measures that are specific for people with dementia could be more sensitive in their assessment of quality of life.

Results from this review indicate that adaptations for people with dementia or MCI were necessary for technology-based interventions to be successful. Guidance was significant, as participants with dementia did not always remember how the technological intervention worked [35]. The need for this adaptation is supported by previously recognized symptoms of dementia, such as memory loss and decline in executive function [62]. Proper safety and preparation for the exercise intervention were recognized as concerns by the researchers. Previous research has identified that persons with MCI or dementia have increased fall risk [63]; thus, additional safety preparation may be warranted. Studies in the review showed that a "one-size-fits-all" game approach did not work; the intervention had to be customizable to the level of the individual participant. Future interventions for individuals with dementia should involve caregiver training for assistance and safety. Clear guidelines are needed for intervention adaptation to meet the needs of specific groups.

Dementia is associated with decreased attention span and other behavioral and psychological symptoms [64]. It is important to note that playing a game that is either too difficult or too easy may cause frustration, agitation, or loss of attention. This is consistent with the individualization and adaptation commonly used for in-person exercise interventions for people with dementia [65]. Future studies should actively involve individuals with dementia and their caregivers in the co-design of technology-based group exercise programs. This inclusive approach ensures that the interventions are tailored to the specific needs and capabilities of the end users [23]. Including caregivers and individuals with dementia in the development process can also identify practical implementation strategies, fostering a sense of ownership and increasing the likelihood of successful adoption in long-term care settings. In the review, participants in one study reported satisfaction with contributing meaningfully to research [17].

Overall, results from the scoping review indicate that technology-based group exercise interventions are feasible for persons with dementia or MCI. These feasibility studies are novel; however, research with similar populations, such as older adults [66] and adults with Parkinson's disease [67], support these findings. Adherence to a technology-based intervention (range of 70–79%) [17, 31] is comparable to adherence to in-person exercise interventions by persons with dementia or MCI (mean of 70%) [68]. Establishing feasibility is important, as there is growing demand in this population for telehealth, technology-based, and remote interventions [69]. Such interventions circumvent current and common issues in accessibility faced by persons with dementia or MCI and their caregivers. For example, need for transportation to site or lack of community resources are common accessibility issues. The convenience and feasibility suggested for further research into the effects and efficacy of such interventions.

## Strengths and limitations

This scoping review presents the first comprehensive overview of multiple types of technology-based group exercise interventions for people with dementia or MCI, building on previous systematic reviews and meta-analyses that analyzed the effects of exergames [70] or virtual reality [71] in people with major cognitive disorders, MCI, or dementia. Although using two researchers to independently extract data followed by discussions among the entire research

team is time-consuming, this methodological approach can effectively reduce the risk of errors and bias, thereby increasing the reliability and validity of the data. Inclusion of family partners and patient partners in conducting the scoping review improved the relevancy of the study by adding their perspectives and lived experiences. Family and patient partners discussed each study with the rest of the research team and were instrumental in extracting the impacts of each study and adaptations necessary for persons with dementia.

This study has limitations. It only considered material referenced as "technology" or other specific known technologies such as "exergame," "virtual reality," "online," and so forth. The field of technology is advancing rapidly and creating new technologies and subsequent new terminologies; "technology-based" is a broad term and may not always be used in favor of more specific terminology. Publications that were not referenced by our search terms may not have been captured. This scoping review did not include literature published in languages other than English. Future research should examine publications and grey literature on technology-based exercise interventions for people with dementia or MCI published in other languages.

## Conclusion

This scoping review suggests that group exercise interventions that utilize technology are feasible and have positive psychosocial, physical and cognitive benefits to persons with cognitive impairment. The use of technology allows for both the intervention and data collection to be conducted remotely, which is essential for maintaining the benefits of group exercise while introducing the convenience of home-based exercise. As technology continues to advance, it is important to investigate inclusive strategies for relevant deveopelment and feasible and sustainable implementation of technology-based group exercise. Future work should explore strategies to involve individuals with dementia and their caregivers in the design and implementation of technology-based group exercise programs.

## Supporting information

**S1 Appendix. Database search strategy for CINAHL.**
(DOCX)

**S1 Table. PRISMA-ScR checklist.**
(DOCX)

## Acknowledgments

The authors acknowledge and thank Senior Medical Librarian Michelle Keba Knecht at Florida Atlantic University Charles E. Schmidt College of Medicine for her assistance. We thank the family and partners of participants for their work and contribution to this scoping review.

## Author Contributions

**Conceptualization:** Lillian Hung, Juyong Park.

**Formal analysis:** Lillian Hung, Hannah Levine, David Call, Diane Celeste, Dierdre Lacativa, Betty Riley, Nathanul Riley, Yong Zhao.

**Funding acquisition:** Juyong Park.

**Writing – original draft:** Hannah Levine, Yong Zhao.

**Writing – review & editing:** Lillian Hung, Juyong Park.

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
