## [Decision Letter · Decision Letter 0]

9 May 2024

PONE-D-24-07416Technology-Based Group Exercise Interventions for People Living with Dementia or Mild Cognitive Impairment: A Scoping ReviewPLOS ONE

Dear Dr. Hung,

Thank you for submitting your manuscript to PLOS ONE. After careful consideration, we feel that it has merit but does not fully meet PLOS ONE’s publication criteria as it currently stands. Therefore, we invite you to submit a revised version of the manuscript that addresses the points raised during the review process.

We look forward to receiving your revised manuscript.

Kind regards,

Yuzhen Xu

Academic Editor

PLOS ONE

Additional Editor Comments:

Please revise the manuscript carefully according to the reviewers' comments.

Reviewers' comments:

Reviewer's Responses to Questions

**Comments to the Author**

1. Is the manuscript technically sound, and do the data support the conclusions?

Reviewer #1: Yes

Reviewer #2: Yes

2. Has the statistical analysis been performed appropriately and rigorously? 

Reviewer #1: N/A

Reviewer #2: N/A

3. Have the authors made all data underlying the findings in their manuscript fully available?

Reviewer #1: Yes

Reviewer #2: No

4. Is the manuscript presented in an intelligible fashion and written in standard English?

Reviewer #1: Yes

Reviewer #2: Yes

5. Review Comments to the Author

Reviewer #1: Interesting paper. Well written in good English language. Methodology elaborated well. results and conclusions drawn are appropriate. Discussion is well written. Limitations mentioned as well. Accept without changes

Reviewer #2: The manuscript was well-written. However, the results section provided little raw data to support the textual narrative of the studies. It is a recommendation to add more tables that can synthesize the studies based on the study objectives. The authors are also suggested to provided the strategy done in the synthesis. It was not clear if the results of the data abstraction were synthesized by a sole individual or discussed amongst the group of authors. This could have an implication in the limitations of the study as this can also shape how data is interpreted.

6. PLOS authors have the option to publish the peer review history of their article (what does this mean?). If published, this will include your full peer review and any attached files.

Reviewer #1: **Yes: **Arsalan Ahmad MBBS, MD (Neurology)

Reviewer #2: No

---

## [Author Response · Author response to Decision Letter 0]

13 May 2024

We are grateful for the insightful comments and suggestions received during the review process, which have significantly enhanced our manuscript. Each comment has been thoroughly addressed, and detailed explanations for the revisions are provided in the accompanying response letter.

Sincerely,

Lillian Hung

---

## [Decision Letter · Decision Letter 1]

28 May 2024

Technology-Based Group Exercise Interventions for People Living with Dementia or Mild Cognitive Impairment: A Scoping Review

PONE-D-24-07416R1

Dear Dr. Hung,

We’re pleased to inform you that your manuscript has been judged scientifically suitable for publication and will be formally accepted for publication once it meets all outstanding technical requirements.

Kind regards,

Yuzhen Xu

Academic Editor

PLOS ONE

Additional Editor Comments (optional):

Reviewers' comments:

Reviewer's Responses to Questions

**Comments to the Author**

1. If the authors have adequately addressed your comments raised in a previous round of review and you feel that this manuscript is now acceptable for publication, you may indicate that here to bypass the “Comments to the Author” section, enter your conflict of interest statement in the “Confidential to Editor” section, and submit your "Accept" recommendation.

Reviewer #2: All comments have been addressed

2. Is the manuscript technically sound, and do the data support the conclusions?

Reviewer #2: Yes

3. Has the statistical analysis been performed appropriately and rigorously? 

Reviewer #2: N/A

4. Have the authors made all data underlying the findings in their manuscript fully available?

Reviewer #2: Yes

5. Is the manuscript presented in an intelligible fashion and written in standard English?

Reviewer #2: Yes

6. Review Comments to the Author

Reviewer #2: (No Response)

7. PLOS authors have the option to publish the peer review history of their article (what does this mean?). If published, this will include your full peer review and any attached files.

Reviewer #2: No

---

## [Editor Report · Acceptance letter]

4 Jun 2024

PONE-D-24-07416R1 

PLOS ONE

Dear Dr. Hung, 

I'm pleased to inform you that your manuscript has been deemed suitable for publication in PLOS ONE. Congratulations! Your manuscript is now being handed over to our production team.

Kind regards, 

on behalf of

Professor Yuzhen Xu 

Academic Editor

PLOS ONE